# Polypropylene Pelvic Mesh: What Went Wrong and What Will Be of the Future?

**DOI:** 10.3390/biomedicines11030741

**Published:** 2023-03-01

**Authors:** Amelia Seifalian, Zeinab Basma, Alex Digesu, Vikram Khullar

**Affiliations:** 1Department of Urogynaecology, Imperial College, London W2 1NY, UK; 2University College London (UCL) Medical School, London WC1E 6DE, UK

**Keywords:** pelvic mesh, polypropylene, pelvic organ prolapse, material, urogynaecology, graphene, Hastalex, smart materials, regenerative medicine, tissue engineering

## Abstract

Background: Polypropylene (PP) pelvic mesh is a synthetic mesh made of PP polymer used to treat pelvic organ prolapse (POP). Its use has become highly controversial due to reports of serious complications. This research critically reviews the current management options for POP and PP mesh as a viable clinical application for the treatment of POP. The safety and suitability of PP material were rigorously studied and critically evaluated, with consideration to the mechanical and chemical properties of PP. We proposed the ideal properties of the ‘perfect’ synthetic pelvic mesh with emerging advanced materials. Methods: We performed a literature review using PubMed/Medline, Embase, Cochrane Library (Wiley) databases, and ClinicalTrials.gov databases, including the relevant keywords: pelvic organ prolapse (POP), polypropylene mesh, synthetic mesh, and mesh complications. Results: The results of this review found that although PP is nontoxic, its physical properties demonstrate a significant mismatch between its viscoelastic properties compared to the surrounding tissue, which is a likely cause of complications. In addition, a lack of integration of PP mesh into surrounding tissue over longer periods of follow up is another risk factor for irreversible complications. Conclusions: PP mesh has caused a rise in reports of complications involving chronic pain and mesh exposure. This is due to the mechanical and physicochemical properties of PP mesh. As a result, PP mesh for the treatment of POP has been banned in multiple countries, currently with no alternative available. We propose the development of a pelvic mesh using advanced materials including emerging graphene-based nanocomposite materials.

## 1. Introduction

The use of transvaginal pelvic mesh for prolapse surgery has become a highly controversial and widely discussed topic among gynaecologists in recent years. When polypropylene (PP) pelvic mesh was first introduced to the healthcare market, it was described to be a simple and permanent tool for treating pelvic organ prolapses (POP). However, there has since been a media flurry and a growing body of complaints from women reporting severe irreversible complications as a result of the procedure. This has resulted in the National Institute for Health and Care Excellence (NICE) and the Food and Drug Administration (FDA) to ban PP mesh for certain prolapse surgeries in the UK and USA, amongst other countries which have followed suit [1]. A timeline has been illustrated to highlight key dates in the regulation and investigations of PP vaginal mesh (Figure 1).

Surgical mesh consists of a flat, net-like structure (Figure 2) that is surgically implanted to support weaknesses of soft tissue, hence its application in treating pelvic prolapses. The mesh creates tension around bodily structures and is thus useful in managing dysfunctions caused by weaknesses of muscle and soft tissue [2]. A number of materials have been used to create the mesh structure for surgical implantation. However, the most common is synthetic PP. Consequential poor patient outcomes are thought to occur either as a result of the type of material used to fabricate the mesh implant or from the procedure performed in order to insert and fixate the mesh implant.

This research aimed to critically investigate the type of materials used for the fabrication of pelvic mesh, how vigorously these materials were tested prior to clinical translation, and identify the key concerns regarding PP material and its properties. We present alternative implants currently in the animal trial stage for preclinical testing with a promising future application in POP surgery.

## 2. Pelvic Organ Prolapse

POP takes place when the pelvic muscles weaken, most often as a result of childbirth and ageing, and so the organs next to or surrounding the vagina descend into the vaginal canal, causing prolapse [3]. POP causes discomfort with symptoms including pain; the feeling of pressure or heaviness ‘down below’; the sensation of a bulge or protruding mass; and a visible bulge or protruding mass coming out of the vaginal canal [4]. The symptoms of POP can have a damaging effect on mental health, social and sexual health, and wellbeing.

In the UK, 8.4% of women report the presence of a vaginal bulge. Based on examination reports, prolapse is present in up to 50% of women, according to NICE [5]. NICE also reports that one in ten women require surgical management for their prolapse and as many as 19% of these women relapse and require further surgery, thereby creating costs for the NHS. The expense to the NHS is further increased with negligence claims which, according to a UK Government House of Commons report, summates to over 800 claims against both the NHS and the companies producing PP mesh implants [6].

Surgical treatment for POP involves strengthening and supporting soft tissue weaknesses with sutures or with the use of a mesh adjunct, as discussed in this research. The clinical transvaginal surgical insertion of PP mesh for POP has been banned in the UK by NICE due to safety concerns. Transabdominal PP mesh surgery is still available and offered to patients for the management of vaginal vault prolapse and uterine prolapse, as a last resort if conservative measures have failed [7]. For both anterior and posterior wall prolapse, NICE has banned all forms of PP mesh surgery, due to the growing body of evidence that PP mesh does not improve symptomatic rates of success, including a cohort study involving nearly 19,000 women undergoing POP surgery [8].

## 3. Material of the Mesh Implants: Synthetic or Biological?

The synthetic PP mesh implant has been the most commonly used for POP surgery due to its easy availability and affordability. However other augmenting materials have also been manufactured and tried, to produce a safer and more suitable implant. The materials used for the fabrication of the mesh implant can be divided into two main categories: synthetic and biological.

Biological grafts provide an alternative to the more popular synthetic mesh and can either be autologous or heterologous. Materials used to develop biological grafts are outlined in Table 1. The cost-effectiveness of biologic grafts is questionable, as they incur greater costs than synthetic materials and studies show they lack clinical effectiveness. Current research demonstrates a high chance of symptomatic relapse and no added benefit compared to native tissue repair [9]. 

The production of autologous mesh grafts requires the harvesting of the patient’s own tissue. The fascia lata or rectus fascia tissue is used most often due to suitability for size and strength [10]. Autologous grafts are more expensive and time-consuming since the patient must undergo two procedures: one procedure to harvest the tissue, followed immediately by the second procedure to insert the implant. Harvesting tissue from the donor site has increased risks for donor site morbidity, including postoperative infection, scarring, nerve injury, and incisional hernia [11]. The successful use of autologous fascia also relies on donor site tissue quality and is therefore not often plausible, as tissue quality worsens with age [12].

Heterologous grafts include harvested materials from allografts, cadavers, and xenografts from animal tissue [13]. The harvested material must undergo processing in order to create an acellular scaffold that integrates with the patient’s tissue. Processing of the grafts is both costly and time consuming. The heterologous tissue that has been harvested using an aseptic technique first needs to be appropriately sterilised with antibiotics and screened for a number of infectious viruses. There have been no cases of viral transmission from heterologous pelvic mesh reported to date, however, the risk needs to be considered and reduced where possible [14]. Decellularisation of the scaffold is required to remove the immunogenic components of the graft and prevent host response. Decellularisation causes weakening of the biomechanical properties of the graft, however, is required, in balance, to prevent an immune response. Natural enzymatic biodegradation of the implant also occurs over time, ultimately resulting in mechanical failure and relapse of symptoms [15]. Heterologous grafts do not offer a permanent solution and are therefore rarely used to treat POP.

A landmark randomised control trial (RCT), PROSPECT, carried out in 2016, studied 1348 female patients undergoing POP surgery to compare the long-term outcomes of native tissue repair versus synthetic or biologic mesh augmentation [16]. This was the largest RCT to date, studying the outcomes of POP surgery with and without mesh. The trial found biologic graft to result in worse outcomes with more prolapse recurrence compared to standard native tissue repair. Similarly, the use of PP mesh also did not affect rates of relapse, although the same study found a complication rate of 12% in patients managed with PP mesh, at the two-year follow up. 

The synthetic meshes include the well-known PP mesh, which dominated the prolapse market. However, the other meshes to note include the polyethylene terephthalate (PET) mesh, Mersilene^®^, and the expanded-polytetrafluoroethylene (ePTFE) soft tissue patch, Gore-Tex^®^. PET is a thermoplastic polymer of the polyester subtype considered to be more inert in vivo compared to PP [17]. Both PET and ePTFE implants are considered to have a greater risk of complications over PP [18]. Ultimately PP was favoured over both PET and PTFE due to having a superior function in triggering tissue ingrowth [19]. Over the years, synthetic mesh has been favoured over alternative biological grafts. The costs of synthetic mesh include production, often cheap, and surgical implantation, a single and quick procedure. Thus, it is considered to be the most time- and cost-effective procedure for POP treatment, whereas the other materials available incur extra costs in harvesting tissue and processing the implant, as well as having a greater risk of relapse. 

## 4. Polypropylene and Its Properties Used as Mesh for Treatment of Pelvic Organ Prolapse

PP is a rigid, strong, and crystalline nonabsorbable thermoplastic produced using the propene monomer, chemical formula (C_3_H_6_)_n_. It is a linear hydrocarbon resin. PP is one of the most widely used polymers for industrial applications to date, due to its low cost and ready availability. It can be used as a plastic or in the form of fibre material. It is nontoxic and thus considered appropriate for use in humans. PP is uniquely useful due to its high electrical and chemical resistance at raised temperatures. Table 2 demonstrates a summary of the mechanical and chemical properties of PP. 

PP is a common material used in the laboratory setting for the manufacturing of plastic test tubes, carboys, and vacuum flasks. PP is also a popular material in other industries, including the manufacturing of furniture, as a result of its strength and high chemical resistance, producing plastic tables and chairs suitable for children. PP is also known to have a number of medical applications, including PP mesh for the surgical repair of hernias as well as in a number of medical devices and investigative equipment.

The development of PP mesh for surgical implantations needs consideration of the biomechanical properties of the material, mesh pore size, stiffness, elasticity, and tissue compatibility. However, in addition, patient characteristics and surgeon technique also affect clinical outcomes [20]. The ideal pelvic implant would have similar biomechanical properties to that of native surrounding tissue. 

Pelvic PP mesh is a flat sheet manufactured by either knitting or weaving a PP fibre. Knitted mesh implants consist of a single filament, looped with itself to create its structure, whereas woven mesh implants consist of two different filaments crossed perpendicularly with each other (Figure 3). Knitted mesh has a higher porosity and is much more flexible than woven mesh. Hence, it is the preferred method of fabrication. Greater flexibility gives the material more resilience, so it can absorb large bursts of energy and deform elastically, providing a longer lifespan in the high-pressure environment of the pelvic cavity. Increased porosity results in a lower mesh burden and less native tissue in contact with the PP material. In turn, this dampens the immune response as less tissue is reacting to the foreign body PP mesh, reducing the risks of treatment failure and complication. The main benefit of PP, over biologic alternatives, is its resistance to enzyme degradation.

PP has both a greater Young’s modulus and greater ultimate tensile strength compared to native pelvic tissue [21]. This means that PP is far more elastic than native pelvic tissue. Therefore, it can withstand greater force and stress without breaking or failing. Amongst other factors, the ideal mesh would need to be able to withstand the changes in pressure in the pelvic cavity when the abdominal contents move during coughing and sneezing. 

High yield strength is important when designing the pelvic mesh implant. This is to prevent plastic deformation of the device once implanted and to maintain efficiency in supporting pelvic organs. A study of PP found that, as a material, it is unable to cope with great strains from pressure, resulting in deformation of the implant and irreversible stretching [22]. Elastic deformation in line with surrounding pelvic tissue is a healthy response to a sudden rise in stress that takes place in the pelvic cavity, as described before. Therefore, elastic deformation is significant with a high upper threshold for plastic deformation.

The soft tissue of the female pelvic anatomy has evolved to cope with high pressure and absorb energy while deforming elastically. Studies report that both high-density and stiffer PP meshes negatively interfere with the contractility of the smooth muscle in the pelvic region [23]. The stiff PP mesh is thought to ‘shield’ the surrounding muscle from the normal physiological forces experienced, and, in turn, accelerates tissue degeneration. This is a phenomenon known as stress shielding. It is known that when a hard material is in contact with a softer material, it results in the erosion of the softer material. Therefore, a mismatch of properties can result in mesh erosion through native tissue. A material that is softer and better matches the properties of native pelvic tissue would prevent the likelihood of mesh erosion [24]. Ultimately the stress-shielding results in vaginal atrophy and hence degrades the quality of soft tissue of the vagina. The vaginal muscles rely on these regular forces that are experienced in order to stretch and maintain their structure and components. 

There are reports of PP mesh causing an adverse immune reaction resulting in inflammation and deposits of fibrotic tissue. Large pore size is an important feature in the design of pelvic mesh. This is to reduce the risk of infection by allowing cell infiltration and integration with the mesh implant [25]. The significance of this is related to the different sizes of the smaller pathogens and larger immune cells [26]. In relevance to pore size, plastic deformation and the natural strain experienced by the mesh, in vivo, results in pore deformation and, ultimately, mesh shrinkage.

The PP pelvic mesh implant is stiffer than native vaginal tissue and, therefore, may be a cause for poor tissue integration and resultant complications. As PP does not have the same viscoelastic properties as native pelvic tissue, during movement it tends to damage the surrounding tissue and does not integrate with the site in which it has been implanted. For the surgical pelvic implant to succeed it is important for the materials to be biocompatible and for the implant itself to have similar biomechanical properties as the surrounding tissue for integration. It is important to note that the presentation of the mesh for use in surgery can also play a role in the development of complications. Transvaginal urethral tapes are usually presented and attached to the introducer and the tape is covered with plastic film, which is removed upon insertion of the tape. This minimises tape microbiome contamination, which has been associated with tape extrusion and complications [27].

## 5. Clinical Complications Arising from the Polypropylene Mesh Material

PP is a hard and heat-resistant plastic, making it well known for medical and industrial applications, as aforementioned. It is a commonly used plastic that is readily available and suitable in a number of markets. PP is ubiquitous in the developed and developing world. With its plentiful industrial applications, it will continue to be produced and manufactured at a high scale for everyday use. 

PP mesh was initially authorised, with FDA, USA, and NICE, UK, approval, for use in the treatment of gastrointestinal hernia repairs, including oesophageal and hiatus hernias. With this original use of PP mesh to repair hernias, no major complications regarding the material were encountered. Therefore FDA clearance for use of the same PP mesh to treat pelvic floor dysfunction was rapid and simple to achieve. Since the PP mesh was rolled out for use to treat POP, new studies have shown long-term severe complications, see Box 1. The most prominent complication is mesh erosion, an issue negligible in the aftercare of gastrointestinal hernia repair and yet causing much controversy in pelvic dysfunction repair [28]. 

Box 1Severe complications arising from the use of polypropylene mesh.
Chronic infectionChronic painDyspareuniaMesh exposure—display of mesh at or near the site of insertionMesh extrusion—where the mesh passes out of a body structurePerforation of neighbouring organs secondary to erosionMesh shrinkageRecurrence of prolapse with treatment failure and further surgery


Testing of vaginal PP mesh, or thence the lack of it, as apparent in the literature and summary of all clinical trials, see Table 3, is described to have been minimal due to an FDA regulation waiver of the standard rigorous testing of products that are based on existing FDA-approved devices. Since PP mesh was already commonly used for hernia repair, thorough testing of the product was not required, under the FDA 510 (k) clearance rule. The FDA 510 (k) pathway provides rapid clearance to products, allowing them to bypass further clinical trials and extra safety regulation processing if they are deemed substantially equivalent to a product already approved and in use in the healthcare market. It is for these reasons that, initially, the long-term implications of pelvic PP mesh may not have been fully known. It is very important to not just evaluate the material, under GMP and GLP practice, but also the evolution of the product made and how the device adapts to the part of the body in which it is implanted.

In order to properly assess the risks and complications of PP mesh, after GMP/GLP preclinical trials, large multicentre clinical studies with long-term post-implantation follow-up are required. Currently, mesh erosion has been described to be the most damaging complication of mesh implant surgery, and it often takes three years or more for such symptoms to arise, hence the importance of long-term follow up in clinical trials. Clinical trials lasting more than three years are ideal as the risk of mesh erosion can be fully assessed to balance the pros and cons of the surgery. The majority of current clinical trials look at early complications and side effects from the surgery, hence why such controversy exists around this subject. It has come to light that there is a lack of clinical trials assessing mesh safety, which has been demonstrated in Table 3.

The main and most controversial complication of mesh insertion is mesh exposure and extrusion, which is caused by synthetic PP mesh, and occurs regardless of location or procedure of insertion. Mesh extrusion (Figure 4) is defined as the exposure of synthetic mesh material through the wall of the vagina and is visible on vaginal examination. Mesh extrusion affects 12% of the patients who have undergone pelvic PP mesh repair surgery in the UK, as reported by NICE in regard to a two-year follow up [29]. It is almost certain that if the follow up were extended to three years or more, the rate of this complication would rise above 12% as the risks of mesh exposure and extrusion increase with time. This is a serious complication due to the side effects it causes that impact the patient’s quality of life, including, though not limited to: vaginal pain, abnormal discharge, vaginal bleeding, and dyspareunia.

Once mesh erosion occurs from the implant of PP mesh, the next mainstay of management is to excise the mesh in order to prevent further damage to adjacent soft tissue. However, the mesh tends to have already integrated with the surrounding soft tissue and so removal requires a number of complex surgeries. Surgeons performing the removal must be trained to do so and be familiar with the procedure so as to reduce the risk of further long-term damage. Mesh removal ultimately results in the recurrence of POP symptoms in around 20% of cases, meaning the patients suffer again from the same conditions with superimposed complications from the initial management [52]. Surgical procedures can vary from partial removal to the complete removal of the PP mesh, which carries high risk for postsurgical vaginal stenosis. Other therapies include more conservative methods such as the use of oestrogens, antiseptics and/or antibiotics to prevent infection and further complications of PP mesh exposure.

## 6. Future Direction

There are several materials being investigated for use in prolapse surgery with few promising candidates having reached in vivo animal trials. This section will only mention the novel devices that have reached preclinical animal trials for the application of pelvic surgery. Polycaprolactone (PCL) has the greatest volume of published research regarding application with animal studies showing promising results [53,54,55,56,57], including further experimental studies combining PCL with stem cells to augment the results [58,59]. Biodegradable PCL has also been included in experimental studies, although results found that degradation of the mesh occurred faster than tissue regeneration, hence, the higher risks of implant failure [60]. Polylactic acid was investigated, in combination with PCL and independently in the rabbit model [61]. Stem cells were also applied to a polyamide mesh in an investigation of its properties to improve integration with surrounding tissue [62].

Polycarbonate has also been investigated in a number of animal studies, including as an isolate material, processed with other chemicals, and for biodegradable purposes—although this is a less appropriate option for long-term anatomical support if tissue regeneration is not successful [63,64,65]. Both polyvinyl plastics, polyvinyl fluoride, and polyvinylidene fluoride were studied and performed using ovine models and a porcine cadaver, respectively [66,67]. Polydimethylsiloxane, a silicone-based organic polymer, has also entered preclinical in vivo animal studies with successful outcomes compared to the standard PP mesh [68].

There remains a gap in the market for a better fitting solution than the current PP mesh with competitive alternatives currently under research. A list of properties that would produce the ideal mesh adjunct has been described, see Box 2. The unmet clinical need has resulted in a race for a novel surgical implant to enter the market of POP surgery.

Box 2The ideal properties of an implant for the augmentation of surgical pelvic organ prolapse repair.
Nontoxic and biocompatibleChemically inertLightweight with low densityLow stiffnessLarge pores and high porosityMechanically strongNondegenerativeNoncarcinogenicNoninflammatory and nonallergenicAffordableSterileResistant to mesh shrinkage


We have been working on the development of vaginal mesh using a graphene-based nanocomposite copolymer [69]. Graphene is a two-dimensional (2D) honeycomb, mono- or multilayer, with sp2 hybridisation. It harbours unique electrical, chemical, optical and mechanical properties. It is 200 times stronger than steel yet at the same time it is known to be extremely elastic and very light [70,71]. We have matched the viscoelastic properties of the mesh, manufactured using Hastalex^®^, to the natural viscoelasticity of pelvic tissue. Hastalex^®^ is unique with it has the properties of superior elasticity and great ultimate tensile strength, see Figure 5. These properties make Hastalex^®^ an ideal candidate for the development of a novel and innovative surgical implant for the management and cure of POP. To enhance the integration of Hastalex^®^ within the pelvic cavity, we have seeded the mesh with stem cells obtained from fat obtained from a surgical procedure. The material is currently undergoing preclinical testing and is under development for application in other medical devices, including a synthetic heart valve [72], muscle tendons, and in the repair and replacement of the human tympanic membrane [73].

## 7. Conclusions

The majority of the clinical trial studies performed proved that PP mesh was biocompatible and nontoxic, though with a risk of complications, most prominently of mesh exposure. Very few studies investigated the mechanical properties of the PP mesh implant and the potential mismatch of viscoelastic properties with surrounding native tissue. The majority of clinical trials were less than three years meaning longer-term complications may have been missed in these studies. The mismatch of viscoelastic properties and the increase in inflammatory processes can cause damage over a long-term period (more than three years) which may result in serious adverse events, hence studies over three years would be most suitable. The ideal mesh should be biocompatible, nontoxic, antibacterial, and have mechanical properties similar to the surrounding pelvic tissue, being able to withstand high pressures without breaking or failing. Currently, a number of advanced materials are under research for surgical application. Polycaprolactone has had the greatest volume of research though other materials also show positive outcomes in preclinical animal trials. Clinical translation of a novel mesh implant would overcome POP, which is currently an unmet clinical condition.

## Figures and Tables

**Figure 1 biomedicines-11-00741-f001:**
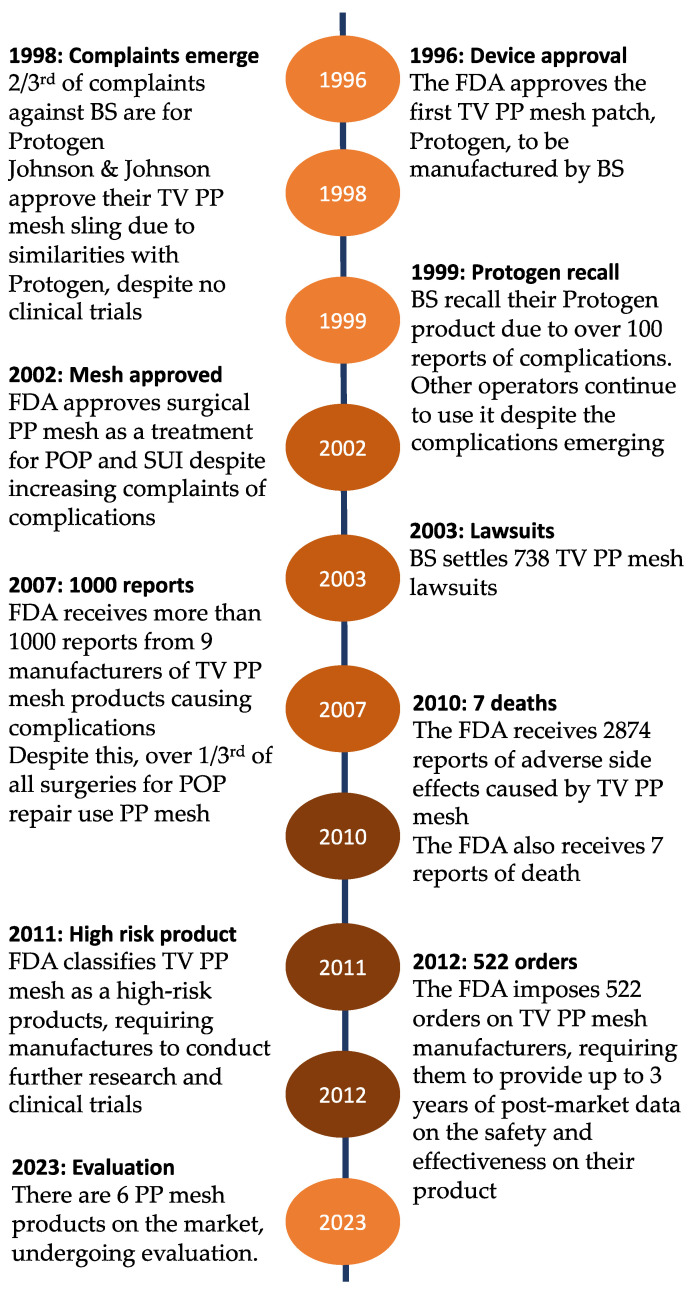
Timeline depicting major events that have taken place since the first introduction of the polypropylene pelvic mesh. Boston Scientific (BS), Polypropylene (PP), transvaginal (TV).

**Figure 2 biomedicines-11-00741-f002:**
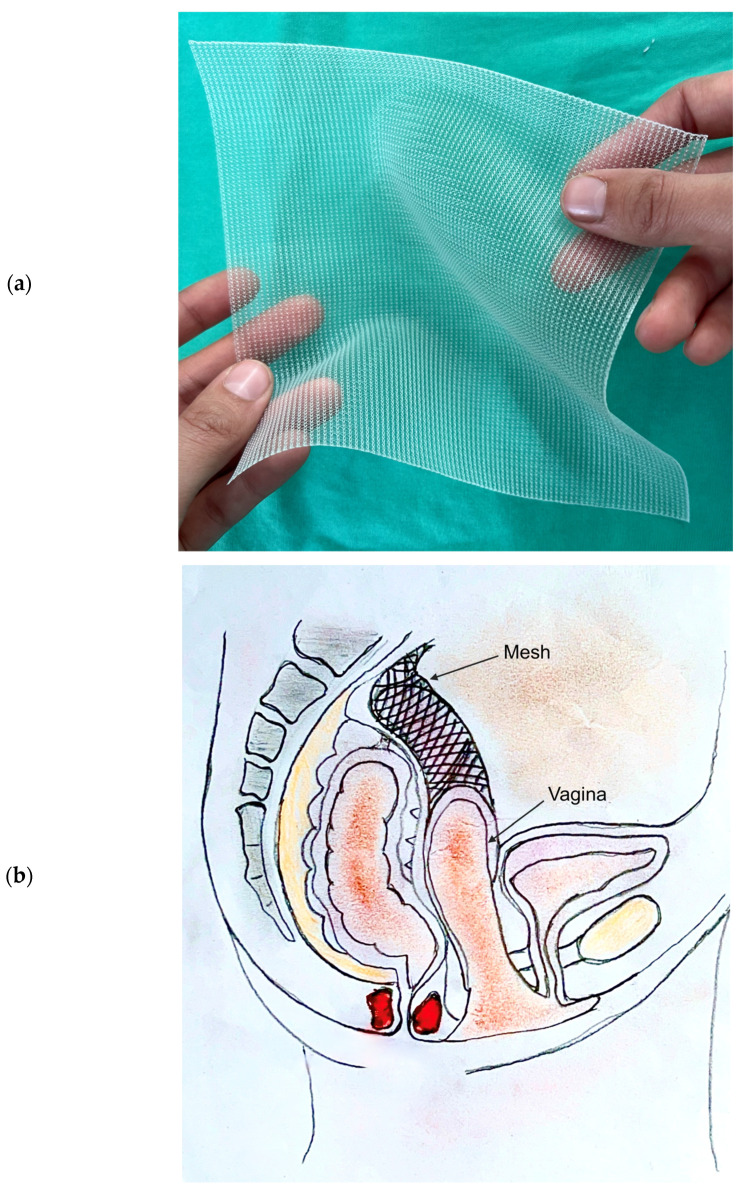
(**a**) Image of polypropylene synthetic mesh used to treat pelvic organ prolapse; (**b**) Image demonstrating the placement of polypropylene mesh in abdominal sacrocolpoplexy—surgical method to treat vaginal vault prolapse.

**Figure 3 biomedicines-11-00741-f003:**
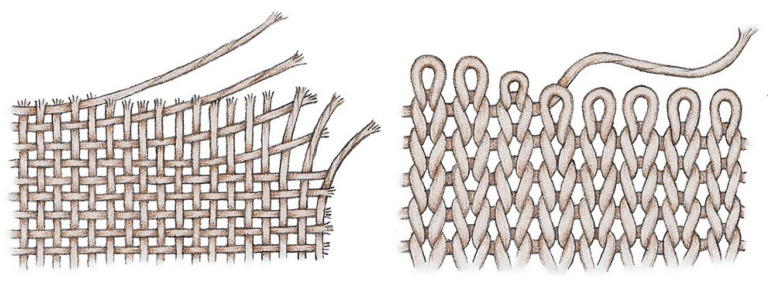
Image showing the structural difference between woven (seen **left**) versus knitted (see **right**) methods of fabrication of the polypropylene mesh, adapted from https://threadden.com/sewing-tips/the-difference-between-knits-wovens/ (accessed on 22 February 2023).

**Figure 4 biomedicines-11-00741-f004:**
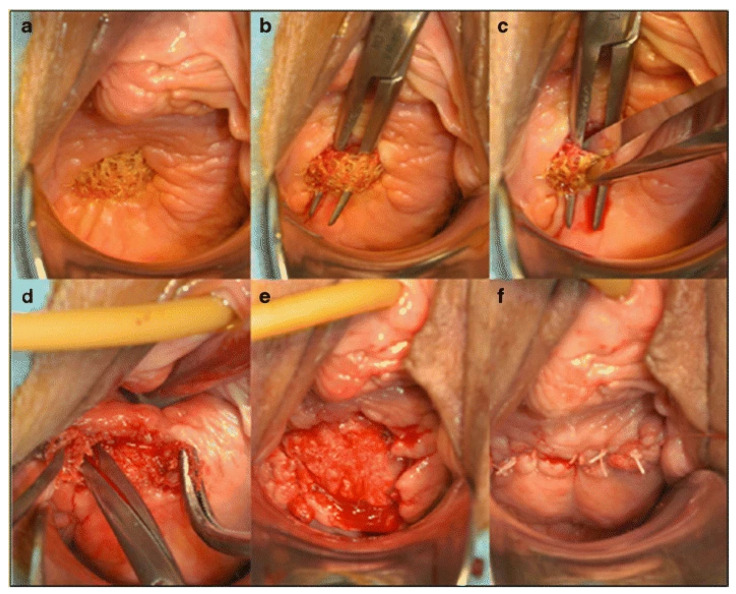
Photograph showing vaginal mesh exposure with steps taken for the partial removal of the mesh. (**a**,**b**) depict the exposure; (**c**) depicts sectioning and removal; (**d**,**e**) dissection of the surrounding tissue; (**f**) depicts wound closure following partial removal of mesh [30].

**Figure 5 biomedicines-11-00741-f005:**
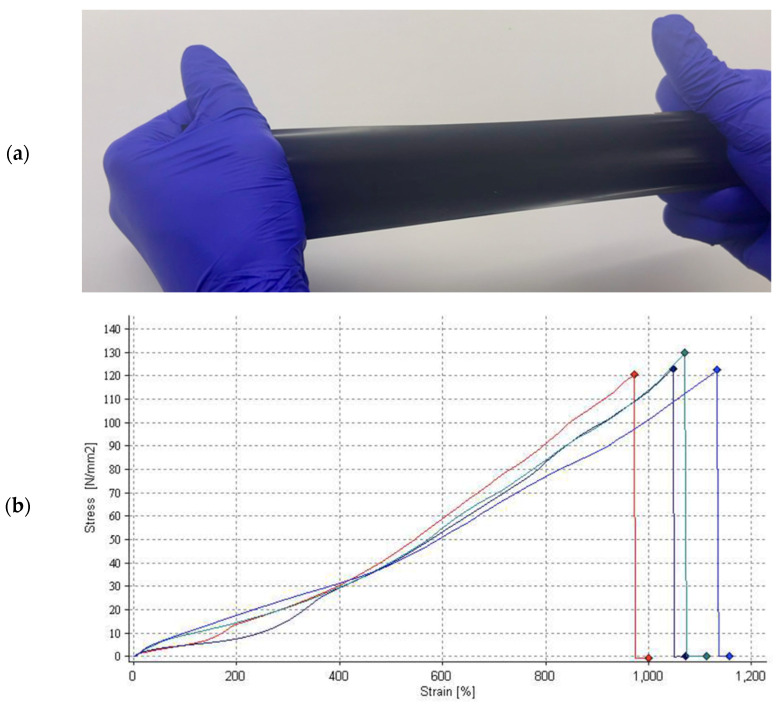
(**a**) Photograph of a sheet of Hastalex^®^, www.goodfellow.com (Accessed on 22 February 2023), (**b**) Hastalex^®^ is a functionalised graphene oxide-based material with high tensile strength and strain. The graph shows four different lines depicting various thickness of material tested. The red line represents the thinnest sample and the green line represents the thickest sample of our novel Hastalex^®^ surgical membrane implant for pelvic organ prolapse.

**Table 1 biomedicines-11-00741-t001:** Materials that have been used clinically to augment pelvic organ prolapse (POP) surgical repair. Disadvantage of the biological materials is their lack of mechanical properties, especially long-term post implantation.

Type	Sub-Type	Material	Advantages	Disadvantages
Biologic	Autologous	Rectus sheath	Most similar to the patient’s own native tissue Unlikely to trigger immune response	Expensive Time-consuming (requires two procedures) Risk of donor site morbidity Further scarring
		Fascia lata		
	Allografts	Cadaveric fascia lata	Reduced likelihood of host response	Lack of mechanical properties Higher rates of treatment failure Greater chances of requiring further surgical procedures for correction following treatment failure
		Cadaveric human dermis		
	Xenografts	Porcine dermis		
		Porcine small intestinal submucosa		
		Bovine pericardium		
Synthetic	Polypropylene	Heavyweight polypropylene	Cheap Easy to produce in mass quantities Easy procedure for insertion with well-designed shapes for ease of insertion	Risk of serious complications including mesh exposure, chronic pain, and chronic infection Difficult to remove in the presence of long-term complications following tissue integration
		Lightweight polypropylene		
	Polyethylene terephthalate	Polyester mesh known as Mersilene^®^	Better cytocompatibility	Smaller pores compared to the polypropylene meshes Greater rate of severe, long-term complications
	Expanded—Polytetrafluoroethylene	Soft tissue patch Gore-Tex^®^	Lower rates of infection and chronic inflammation	More likely to result in treatment failure, requiring further surgery for relapse

**Table 2 biomedicines-11-00741-t002:** Summary of the mechanical and chemical properties of polypropylene material in its standard form. Note that the method of fabrication can significantly affect material properties.

Mechanical Properties	Chemical Properties
Low density (0.9 g/cm^3^) Lighter than water Floats	Great chemical resistance Resistant to most organic solvents at room temperature
Tough Acts with elasticity over a range of deflection to prevent deformation	Highly resistant to dilute or concentrated acids, alcohols, and bases
Flexible Semi-crystalline nature gives it a high flexural strength	Highly impermeable to water
High resistance to fatigue Able to retain shape after a large degree of torsion and bending	High electrical resistance
High tensile strength Around 4800 psi Able to withstand heavy loads despite being lightweight	Thermoplastic Becomes liquid at the melting point Can be melted, cooled, and reheated again without degradation
High impact tolerance	
Hard consistency Semi-rigid structure makes it more likely to bend and flex with impact	

**Table 3 biomedicines-11-00741-t003:** Summary of Clinical Trials, www.clinicaltrials.gov (accessed on 22 February 2023), of polypropylene (PP) vaginal mesh. Keys: FU, follow up in month; OS, ongoing study; POP, pelvic organ prolapse; SUI, stress urinary incontinence; ME, mesh exposure; FE, Faecal incontinence; Pt, patient; RCT Randomised controlled trial; SFA, Sexual Functional Abnormality; TVM, transvaginal mesh, * indicate study incomplete.

Material	Pt Demographic	Country	Study Method	FU	Complications	Outcome	Year, Ref
A low-weight PP mesh	202 with POP	Finland	A prospective RCT comparing low-weight PP mesh for anterior vaginal wall prolapse vs. traditional anterior colporrhaphy.	24	ME was seen in 8% of Pts. The dyspareunia score was lower in the mesh group.	Overall symptom rates did not differ in mesh and nonmesh groups. Recurrence rates were 11% in mesh group vs. 41% in nonmesh group.	2003 [31]
Pelivisoft—natural organic mesh made of porcine dermis vs. Pelvitex—PP mesh	120 with POP	US	An RCT to compare the relative differences between the materials Pelivisoft vs. Pelvitex, as an adjunct to sacral colpopexy surgery.	12	One Pt in the Pelvisoft group experienced ME. Two Pts in the porcine group, compared to three Pts in the synthetic mesh group, experienced dyspareunia.	Anatomic cure rates for the Pelvisoft and Pelvitex groups were 81% and 86%, respectively. “Clinical cure” rates for the Pelvisoft and Pelvitex groups were 84% and 90%, respectively.	2005 [32]
Polyform—PP mesh developed by Boston Scientific	100 with POP	US	A prospective single-blind RCT, to evaluate anterior colporrhaphy vs. cystocele repair using PP mesh or porcine dermis.	24	ME in 14% of Pts in the mesh group. Composite failure was 4% in the mesh group, 12% in porcine and 13% in colporrhaphy group.	Pts in PP mesh group had a significantly lower anatomic failure rate (18%) than the porcine (46%) and colporrhaphy groups (58%).	2005 [33]
Ugytex—low-weight and highly porous PP monofilament mesh	194 with POP	France	A multicentre study to evaluate and compare the prosthesis Ugytex via the transobturator approach vs. anterior colporrhaphy for the surgical treatment of anterior vaginal wall prolapse.	84	ME in 13% of Pts during FU. Reintervention for prolapse took place in 9% of Pts.	Similar functional outcomes were seen for both mesh and native tissue repair. Use of mesh did not reduce repeat surgery rates but did reduce anatomical recurrence.	2005 [34]
PROLIFT^®^ system—macroporous PP TVM using a transobturator or transgluteal approach	260 with POP	Sweden	A prospective multicentre open labelled single cohort study to describe perioperative complications after TVM surgery for POP.	6	Serious complications were seen in 4% of Pts. Plus, an extra 15% of Pts experienced minor complications.	Perioperative serious complications are uncommon in the majority of cases after TVM procedure.	2006 [35]
NAZCA TC™ POP repair system—macroporous PP mesh	104 with POP	Brazil	An RCT to compare colporrhaphy vs. NAZCA TC™ (Promedon HQ, General Manuel Savio L3 M3, Parque Industrial Ferreyra, X5123XAD, Córdoba, Argentina) for the surgical treatment of greater anterior vaginal wall prolapse.	12	ME was seen in 5.7% of Pts who underwent surgery with PP mesh adjunct.	Monoprosthesis with combined pre-pubic and trans-obturator arms had high success rates for anterior vaginal POP repair and simultaneous SUI treatment.	2007 [36]
Tension-free vaginal tape PP mesh and transobturator tape PP mesh and suprapubic sling PP mesh	92,246 with SUI	UK	A retrospective cohort study using hospital episode statistics data measuring complications following tension-free vaginal tape, transobturator tape and suprapubic sling procedures for SUI.	96	Peri-procedural and 30-day complication rates were 2.4% and 1.7%, respectively; 6% were readmitted at least once within 5 years.	Study shows that estimated of 9.8% of women experience complications either peri-procedural, within 30 days or within five years of surgery.	2007 [37]
Synthetic monofilament PP mesh	65 with POP	USA	A double-blind RCT testing the hypothesis that the addition of a standardised technique of inter-positional synthetic PP mesh placement improves the one-year outcome of vaginal reconstructive surgery for POP vs. traditional vaginal reconstructive surgery.	36	The study was prematurely halted once a ME complication rate of 16% had been met.	There was no difference in three-year cure rates when comparing Pts undergoing traditional vaginal prolapse surgery without mesh vs. those undergoing vaginal colpopexy repair with mesh.	2007 [38]
GYNECARE PROLIFT + M * Pelvic Floor Repair System—a new lightweight PP mesh	127 with POP	Belgium	A prospective multicentre study to evaluate the clinical performance of the GYNECARE PROLIFT + M * Pelvic Floor Repair System for the repair of vaginal POP.	12	ME rate was 10.2% and rate of de novo dyspareunia was 2%.	86% of Pts indicated their prolapse situation to be “much better” following surgery.	2008 [39]
Midurethral tension-free vaginal tape—macroporous PP mesh	160 with SUI	Switzerland	A prospective RCT comparing retropubic transvaginal tape (TVT) with the transobturator tape (outside-in TOT or inside-out TVT-O) sling operation in the treatment of female SUI or stress dominated mixed urinary incontinence.	12	Five ME complications identified. 2% TVT, 17% TOT, and 0% TVT-O Pts reported de novo sexual dysfunction, considered significant enough to halt the study.	There was no difference for Q-max between TVT, TOT and TVT-O. Female sexual dysfunction and ME may be higher with a transobturator tape.	2008 [40]
Avaulta—collagen-coated prolene mesh	138 with POP	Norway, Sweden, Finland and Denmark	An RCT comparing conventional anterior colporrhaphy vs. surgery with Avaulta.	36	ME occurred in 13% of Pts at one-year FU, this number did not change by three-year FU.	The objective outcome was superior in the mesh group, but the use of mesh had no impact on the subjective outcome.	2008 [41]
Alyte©—monofilament type 1 PP lightweight Y-mesh	150 with POP	USA	A prospective study looking at the outcome of Pts who underwent a robotic approach to sacral colpopexy using a PP mesh.	12	Nil noted.	Robotic sacrocolpopexy using Alyte© offers excellent subjective and objective results, the clinical cure rate was 95%, and the objective anatomic cure rate was 84%.	2009 [42]
Rectangular PP mesh vs. one posterior rectangular and one anterior PP Y-mesh	72 with POP	Italy	A prospective RCT to evaluate the outcomes of hysterocolposacropexy with one posterior rectangular and one anterior PP Y-mesh vs. colposacropexy with two rectangular meshes.	51	Recurrent low-grade cystoceles developed in 2.6% and 14.7% of Pts and low-grade rectoceles in 15.8% and 8.8% of Pts in the colposacropexy and hysterocolposacropexy groups, respectively.	Whether the uterus was preserved or not, Pts had similar results in terms of prolapse resolution, urodynamic outcomes, improvements in voiding and sexual dysfunctions.	2010 [43]
PP mesh vs. biological graft	232 with POP	China	A single-blind randomised controlled prospective study evaluating the efficacy, quality of life and complication rates of PP mesh vs. biological graft.	12	Adverse events occurred with significantly different frequencies over 1 year.	Similar recurrence rates for PP mesh vs. biological grafts, at short-term FU. Eating soy products often and vaginal intercourse after surgery reduced recurrence.	2010 [44]
Vypromesh^®^ vs. Ultrapromesh^®^ vs. Prolene light mesh^®^	144 with SUI	Turkey	A prospective RCT evaluating the effectiveness and complications of three types of synthetic mesh materials in sling surgery.	48	4% of Vypromesh^®^, 2% of Ultrapro^®^ and 4% of Prolene light mesh^®^ Pts experienced ME, respectively.	Ultrapro^®^ mesh can be used in sling surgery owing to its higher success rates, and lower rates of ME and de novo urgency rates, as shown in clinical studies.	2011 [45]
Ajust^®^ sling vs. standard mid-urethral slings	419 with SUI	Denmark, Norway and Sweden	An RCT (without blinding) investigating the Ajust^®^ system vs. the current standard mid-urethral sling.	36	There were no major complications in either group. Minor complications of urinary tract infections were noted in both groups.	Ajust^®^ appears equally as safe and effective as the standard mid-urethral sling with regards to long-term FU of Pt-reported outcomes.	2012 [46]
Four-arm PP TVM	160 with POP	Poland	An interventional clinical trial to study the safety and efficacy of performing modified anterior TVM surgery using a four-arm PP mesh adjunct for the treatment of advanced urogenital prolapse after hysterectomy.	24	Intraoperative bladder injury in 4% of Pts. 3% complained of de novo SUI. Vaginal vault prolapse recurred in 6% of cases. ME seen in 1% of Pts at six months.	Four-arm PP TVM is safe and effective and provides an alternative treatment option for vaginal vault prolapse, especially in Pts with contraindications to laparotomy and laparoscopy.	2014 [47]
Upsylon™ (Boston Scientific, 300 Boston Scientific Way, Marlborough, MA 01752-1234, USA) —light-weight PP mesh using permanent sutures vs. delayed absorbable sutures for attachment	198 with POP	US	An RCT to compare mesh and suture exposure rates in women undergoing robotic total hysterectomy and sacrocolpopexy with mesh adjunct with attachment using permanent sutures vs. delayed absorbable sutures.	12	Mesh/suture exposure rate of 5.1% in surgery with permanent sutures vs. 7% in surgery with delayed absorbable sutures. 3% of women experienced a serious adverse event.	Suture type used (permanent or delayed absorbable) for vaginal graft attachment did not significantly exposure rates.	2015 [48]
Bilateral abdominal sacral hysteropexy with Prolen^®^—PP mesh	22 with POP	Turkey	A single-blind RCT investigating anatomic and sexual outcomes of bilateral sacral hysteropexy with Prolen^®^.	18	Too few Pts to evaluate complications.	This technique appears to provide an adequate clinical resolution, and it has potential to be the primary surgical option for women with POP.	2015 [49]
TiLOOP^®^ PRO A-titanised PP mesh (alloplastic mesh)	54 with POP	Germany	A multicentre nonrandomised observational clinical investigation to determine usability and collect post-market information on the TiLOOP^®^ PRO A anterior pelvic floor reconstruction meshes, and to determine its effect on quality of life.	12	No adverse events related to the investigational device.	Positive outcomes achieved in the reconstruction of the anatomical position of the pelvic floor organs. Pts benefit from anatomical stability as well as improved quality of life, with justifiable risks.	2016 [50]
Ethicon J&J Prolene mesh	15 with POP	Turkey	An observational study assessing cases of Pts with POP undergoing laparoscopic lateral suspension with mesh and anatomic success measured using transperineal ultrasonography.	12	OS	OS	2016 *
Synthetic PP mesh vs. synthetic sutures	358 with POP	Canada	An interventional randomised controlled multicentre trial to compare experimental bilateral sacrospinous vaginal vault fixation with synthetic mesh arms vs. standard sacrospinous ligament suspension with synthetic sutures.	24	OS	OS	2016 *
Two rectangular PP meshes using four absorbable sutures with a nonabsorbable 0.0 PP suture	100 with POP	Italy	A perspective randomised trial comparing laparoscopic sacrocolpopexy (LASC) vs. robotic-assisted colposacropexy (RASC) for POP repair, both using PP mesh adjunct. In cases of uterus preservation, the anterior mesh was Y-shaped.	24	ME rate of 4% in the RASC and 6% in LASC group, all asymptomatic and managed expectantly.	RASC provides outcomes as good as those of LASC with 100% anatomic correction of the apical compartment. RASC was somewhat more efficient and associated with fewer cases of persistent prolapse.	2018 [51]
Ethicon J&J PP mesh	52 with POP	Egypt	An RCT comparing lateral suspension vs. sacropexy for the treatment of apical POP.	18	OS	OS	2019 *

## Data Availability

Data sharing is not applicable. No new data were created or analysed in this study. Data sharing is not applicable to this article.

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
