# Peer review of "Polypropylene Pelvic Mesh: What Went Wrong and What Will Be of the Future?"

_biomedicines, 2023, doi:10.3390/biomedicines11030741_

Round 1

Reviewer 1 Report

The manuscript entitled "Polypropylene Pelvic Mesh: What went wrong and what will be of the future?" provides critical insights into using polypropylene mesh. The authors discussed various research and case studies in this review article. However, the major drawback of this review article is that it does not include current research in the field. General comments are as follows:

1. Figure 1: authors are recommended to increase the font size of the text to impart better readability.

2. Few research studies use other synthetic materials as well. A comparative table with polypropylene will greatly improve this manuscript.

3. Relevance of the figure 4 is not clear in this manuscript. please explain

Author Response

We would like to thank the reviewer for his/her comments. We have revised the manuscript accordingly. The aim of this paper was to critically review the use of  polypropylene for the pelvic mesh and its failure in long-term clinical use. However, we have expanded on Table 1 to include comparative features and included further discussion in section 3 on alternative synthetic materials. Please see section 6, which also includes alternatives to polypropylene synthetic mesh which are currently under development. Figure 1 has been edited for better readability. Figure 4 has been removed from the manuscript as has been recommended.

Reviewer 2 Report

The current work by Seifalian et al, reviewed the current management options for pelvic organ prolapse (POP) and if polypropylene mesh a viable option for the treatment of POP. The review provides a good and timely account of the topic. A few points to consider are as follows:

Make sure that the figures have been taken from properly reviewed sources with authentic references.

A paragraph/section on existing alternatives to PP mesh should be included as well.

Include some prominent findings/results (figure) of graphene-based nanocomposite materials.

A Table illustrating the problems with PP mesh should be provided including points such as Mesh erosion, pain, revision surgery, Infection, urinary problems, and recurrence.

In Table 1, the advantages, disadvantages and references for each should be included. In addition to this list, other absorbable synthetic polymers such as polyglycolic acid or polycaprolactone and composite materials could also be incorporated.

Author Response

We would like to thank the reviewer for their valuable comments, we appreciate the time and effort spent on our paper. We have revised the paper according to the comments and believe it has improved significantly.

We removed figure 4 from the paper as recommended. We have made appropriate changes to the remaining figures, some of which were self-obtained.

We have outlined the alternative materials used for POP surgery, with detailed discussion and references in section 3. We have added a summary of the complication of polypropylene mesh, please see Box 1.

We have included a figure of the graphene-based nanocomposite materials and their superior mechanical properties, please see figure 5.

Table 1 now includes the most commonly used materials to augment prolapse surgery. We discuss these in further detail including the advantages and disadvantages of each material. Section 6 includes materials currently under investigation as alternatives to polypropylene mesh.

Reviewer 3 Report

This manuscript entitled “Polypropylene Pelvic Mesh: What went wrong and what will be of the future?” has critically reviewed the current management options for POP and PP mesh as a viable clinical application for the treatment of POP. Based on the safety and suitability of PP material were rigorously studied and critically evaluated, with consideration to the mechanical and chemical properties of PP. The authors proposed the ideal properties of the perfect synthetic pelvic mesh with emerging advanced materials. The authors did a good job, and the review is worthy of publication. But some minor issues need to be solved before publication. For example, the words starting with line 95 on page 4 do not need italics; the comma after Figure 4 on page 8, line 227 needs to be deleted.

Author Response

We would like to thank the reviewer for their comments. We have revised the manuscript accordingly.

Round 2

Reviewer 1 Report

The authors addressed all the queries raised. This manuscript can now be accepted in the present form.